# Polysaccharide-Stabilized Semisolid Emulsion with Vegetable Oils for Skin Wound Healing: Impact of Composition on Physicochemical and Biological Properties

**DOI:** 10.3390/pharmaceutics16111426

**Published:** 2024-11-08

**Authors:** Giovanna Araujo de Morais Trindade, Laiene Antunes Alves, Raul Edison Luna Lazo, Kamila Gabrieli Dallabrida, Jéssica Brandão Reolon, Juliana Sartori Bonini, Karine Campos Nunes, Francielle Pelegrin Garcia, Celso Vataru Nakamura, Fabiane Gomes de Moraes Rego, Roberto Pontarolo, Marcel Henrique Marcondes Sari, Luana Mota Ferreira

**Affiliations:** 1Center for Studies in Biopharmacy, Pos-Graduate Program in Pharmaceutical Sciences, Department of Pharmacy, Federal University of Paraná, Curitiba 80210-170, PR, Brazil; giovannaaraujo@ufpr.br (G.A.d.M.T.); laienealves@ufpr.br (L.A.A.); raulunalazo@gmail.com (R.E.L.L.); pontarolo@ufpr.br (R.P.); 2Department of Pharmacy, Midwestern State University, Guarapuava 85040-167, PR, Brazil; kadallabrida@gmail.com (K.G.D.); jessica_breolon@yahoo.com.br (J.B.R.); juliana.bonini@gmail.com (J.S.B.); 3Laboratory of Technological Innovation in the Development of Pharmaceuticals and Cosmetics, Department of Basic Health Sciences, State University of Maringá, Maringá 87020-900, PR, Brazil; kaahnunes07@gmail.com (K.C.N.); fpgarcia2@uem.br (F.P.G.); cvnakamura@uem.br (C.V.N.); 4Department of Clinical Analysis, Federal University of Paraná, Curitiba 80210-170, PR, Brazil; rego@ufpr.br

**Keywords:** emulsion, avocado oil, blackcurrant oil, natural gums, vegetable oils

## Abstract

**Background/Objectives:** The demand for natural-based formulations in chronic wound care has increased, driven by the need for biocompatible, safe, and effective treatments. Natural polysaccharide-based emulsions enriched with vegetable oils present promising benefits for skin repair, offering structural support and protective barriers suitable for sensitive wound environments. This study aimed to develop and evaluate semisolid polysaccharide-based emulsions for wound healing, incorporating avocado (*Persea gratissima*) and blackcurrant (*Ribes nigrum*) oils (AO and BO, respectively). Both gellan gum (GG) and *kappa*-carrageenan (KC) were used as stabilizers due to their biocompatibility and gel-forming abilities. **Methods:** Four formulations were prepared (F1-GG-AO; F2-KC-AO; F3-GG-BO; F4-KC-BO) and evaluated for physicochemical properties, spreadability, rheology, antioxidant activity, occlusive and bioadhesion potential, biocompatibility, and wound healing efficacy using an in vitro scratch assay. **Results:** The pH values (4.74–5.06) were suitable for skin application, and FTIR confirmed excipient compatibility. The formulations showed reduced occlusive potential, pseudoplastic behavior with thixotropy, and adequate spreadability (7.13–8.47 mm^2^/g). Lower bioadhesion indicated ease of application and removal, enhancing user comfort. Formulations stabilized with KC exhibited superior antioxidant activity (DPPH scavenging) and fibroblast biocompatibility (CC_50%_ 390–589 µg/mL) and were non-hemolytic. Both F2-KC-AO and F4-KC-BO significantly improved in vitro wound healing by promoting cell migration compared to other formulations. **Conclusions:** These findings underscore the potential of these emulsions for effective wound treatment, providing a foundation for developing skin care products that harness the therapeutic properties of polysaccharides and plant oils in a natural approach to wound care.

## 1. Introduction

The process of cutaneous wound healing is intricate and dynamic, involving a series of coordinated biological events such as hemostasis, inflammation, cellular proliferation, and tissue remodeling [1,2]. Maintaining sterility, reducing oxidative stress, and controlling moisture levels are essential to promoting effective wound healing. This creates an optimal environment for fibroblast recruitment and collagen formation [3]. These conditions can be rapidly and effectively achieved through cutaneous formulations, enhancing acute and chronic wound management [4,5,6]. There has been a growing interest in using natural polymers, such as polysaccharides, to formulate skincare products [5,7,8] given their biocompatible properties and potential to create moisturizing gels that can protect the wound area and promote healing [6,7,8]. Polysaccharides, such as gellan gum (GG) and *kappa*-carrageenan (KC), have been broadly studied due to their gelling properties [7,8,9], in addition to their potential to stabilize cutaneous-friendly semisolid emulsions [7].

In this trend of natural excipients, vegetable oils are well-known materials in the context of skin wound treatment. The multiple biologically active compounds, such as linoleic acid, omega-3, and omega-6, promote accelerated skin lipid barrier regeneration [10]. Vegetable oils have demonstrated significant potential in enhancing the wound healing process due to their anti-inflammatory, antioxidant, and antimicrobial properties and ability to support skin repair mechanisms. Studies have shown that certain vegetable oils can modulate inflammatory responses, reduce oxidative stress, and promote fibroblast proliferation, all contributing to accelerated wound healing and improved skin regeneration [11]. The fatty acids present in oils, such as oleic, linoleic, and linolenic acids, are particularly beneficial, as they play a key role in restoring the skin’s lipid barrier and supporting cellular functions critical to wound repair [12]. Research on oil blends, including formulations containing sunflower, canola, and linseed oils, has highlighted their ability to enhance wound closure rates and regulate the release of cytokines involved in inflammation, facilitating a more efficient healing process [13]. Additionally, oils with higher ratios of polyunsaturated fatty acids, such as those found in avocado and blackcurrant oils, have shown superior lipid barrier repair effects and support for collagen synthesis, essential for tissue remodeling in wound care [10,14]. The evidence suggests that the balanced composition of fatty acids from natural oils can offer a viable and effective alternative for wound treatment, combining biocompatibility with multifunctional therapeutic effects.

Avocado oil (*Persea americana* Mill; AO) and blackcurrant oil (*Ribes nigrum*; BO) are rich in essential fatty acids, vitamins, and antioxidants, which are known to promote wound healing. Avocado oil is particularly recognized for its anti-inflammatory and antioxidant properties, which help reduce inflammation and protect skin cells from oxidative damage [10]. Similarly, BO contains high levels of polyunsaturated fatty acids that contribute to maintaining the skin barrier integrity and promoting tissue regeneration [15].

There is consistent evidence indicating that the combination of natural excipients, such as polysaccharides and vegetable oils, may offer a synergistic approach to wound care [16,17]. Incorporating oily materials into topical formulations is challenging due to physicochemical restrictions, such as their hydrophobic nature, which hinders mixing with aqueous components and reduces formulation stability [18]. Using polysaccharides as emulsifying agents could overcome these challenges by creating stable emulsions [19]. Furthermore, the formation of a gel–cream matrix can provide a protective barrier on the skin surface, helping to maintain hydration and protect the wound area from infection [20].

Given the growing demand for effective, safe, and sustainable wound care solutions, the exploration of biocompatible and biodegradable materials like natural polysaccharides and vegetable oils offers a promising pathway for developing advanced therapeutic formulations. Polysaccharides not only exhibit desirable gelling properties, but also serve as stable, hydrating matrices that can be applied directly to the skin, creating a protective barrier conducive to healing. Additionally, vegetable oils are rich in essential fatty acids, antioxidants, and anti-inflammatory agents that address critical aspects of wound healing by promoting cellular repair, collagen synthesis, and oxidative stress reduction. By combining these natural excipients, the present study aims to create an innovative emulsion that leverages the synergistic effects of polysaccharides and vegetable oils, providing a bioactive and structurally supportive platform for wound management. This approach addresses current limitations in wound care by focusing on the development of formulations that are not only effective and multifunctional, but also align with the growing emphasis on environmentally conscious and biocompatible pharmaceutical applications.

Therefore, in this study, we developed and characterized semisolid emulsions stabilized with GG and KC to incorporate AO or BO and enable their use as a cutaneous wound platform. The formulations’ physical stability, rheological properties, and bioadhesion potential were assessed. In addition, biocompatibility was estimated through cytotoxicity assays using the L-929 fibroblast cell line and hemolysis test. Lastly, we examined the healing potential of the semisolids in an in vitro wound model.

## 2. Materials and Methods

### 2.1. Materials

The low acyl gellan gum (GG) and *kappa*-carrageenan (KC) were donated by CP Kelco (Limeira, Brazil). Cetostearyl alcohol, cetostearyl alcohol ethoxylated, and avocado oil (AO) were purchased from Delaware (Porto Alegre, Brazil). Blackcurrant oil (BO) was purchased from Laszlo (Belo Horizonte, Brazil). Vitamin E and Nipaguard^®^ are manufactured and distributed by Engenharia das Essências Ltd.a (São Paulo, Brazil). EDTA (Ethylenediamine tetraacetic acid) and Disodium Salt Dihydrate P.A (For Analysis) was purchased from Neon (São Paulo, Brazil) and propylene glycol P.A (for analysis) from VETEC (Rio de Janeiro, Brazil). 1-1-diphenyl-2-picrylhydrazyl (DPPH) radical and 3(4,5-dimethyl)-2,5diphenyl tetrazolium bromide (MTT) were purchased from Sigma-Aldrich Co. (San Luis, MO, USA). Dulbecco’s modified Eagle’s medium (DMEM), penicillin/streptomycin, and fetal bovine serum (FBS) were obtained from Gibco (Baltimore, MD, USA).

### 2.2. Polysaccharide-Based Semisolid Emulsion

The formulations were produced by emulsifying two phases (Table 1 and Figure 1) [21,22]. Cetostearyl alcohol ethoxylate and cetostearyl alcohol are essential components of the emulsification system. Cetostearyl alcohol ethoxylate, a non-ionic oil-in-water emulsifier with an HLB value of 12–15, reduces interfacial tension for emulsion stability. Cetostearyl alcohol, with an HLB of around 5–6, acts as a co-emulsifier and thickening agent, enhancing viscosity and texture. Both agents ensure product uniformity, spreadability, and stability, making them suitable for effective wound healing applications. The components of the oily phase were heated to 70 °C until melted, while the aqueous phase was heated to 75 °C using heating plates (HJ-3(ATRA) model, Ionlab, Araucária, Brazil). Both KC or GG were added to the heated aqueous phase at 75 °C for complete dissolution and hydration. This step is essential, as adequate heat and mixing help form a gel-like matrix that enhances the stability and viscosity of the emulsion. Once the components were fully dissolved, the aqueous phase was added to the oily phase with constant manual stirring under a low speed until the mixture reached room temperature (25 °C). The manual stirring was maintained for approximately 15 min to ensure proper integration, resulting in a smooth and homogeneous texture. The selected oils, AO and BO, along with Nipaguard^®^, were added only after the mixture had cooled to room temperature. This choice was made to preserve the bioactive properties of these oils, which could degrade if exposed to high temperatures during the emulsification process. Adding oils at a lower temperature helps retain their beneficial components, such as unsaturated fatty acids, vitamins, and antioxidants, which are critical for the intended wound healing effects of the formulations.

The final weight of the formulations (10 g) was adjusted with water, and they were stored in plastic containers at room temperature. Additionally, placebo formulations, which did not contain AO or BO, were prepared for comparison (Table 2).

Table 2 shows the names and compositions of polysaccharides and oils in semisolid emulsions.

### 2.3. Macro and Microscopic Evaluation

The formulations were evaluated regarding macroscopic properties, such as color, odor, and the presence of sediments or indications of instability. For microscopic analysis, a small aliquot of the emulsion was placed on a clean glass slide, covered with slips to create a thin layer for optimal viewing. Microscopic images were obtained using an optical PrimoStar microscope (Carl Zeiss, Jena, Germany) at a magnification of 10× to identify the presence of crystals or oily droplets.

### 2.4. pH

A 10% water dispersion of the formulations was prepared to assess pH values using a previously calibrated potentiometer (PG-1800 model, Gehaka, São Paulo, Brazil).

### 2.5. Density

An approximate weight of 10 g was measured for each formulation, and the volume occupied in a test tube was subsequently determined [21]. The calculation was performed using Equation (1):d (mg/mL) = m/V,(1)
where d is density, m is the exact mass of semisolid weighted, and V is the volume occupied by the mass.

### 2.6. Physical Stability

To assess the physical stability, the formulations were submitted to the centrifugation test [21]. Approximately 10 g of each emulsion was placed in plastic tubes and centrifuged using a centrifuge (Centribio 80-2B model; Biosystem, Campinas, Brazil) at a rotation speed of 3500 rpm for 15 min. Following the test, the semisolids were visually inspected for any instability phenomena, such as phase separation.

### 2.7. Fourier Transform Infrared (FTIR) Spectroscopy

Fourier transform infrared (FTIR) spectroscopy was used to determine the chemical composition and interactions of the semisolids. The formulation and raw materials samples were examined using a Bruker FTIR spectrometer (Karlsruhe, Germany), specifically the Alpha-T FTIR model equipped with an Attenuated Total Reflectance (ATR) module. The spectra of 24 scans, ranging from 4000 to 400 cm^−1^, were used with a resolution of 4 cm^−1^ [23,24].

### 2.8. Principal Component Analysis (PCA)

Following acquiring the ATR-FTIR spectra, we applied principal component analysis (PCA) to delineate sample groups, pinpoint relevant variables, and identify potential outliers. Preprocessing the raw data constituted a crucial preliminary step, significantly enhancing the data quality employed in chemometric analyses. The transformation of the raw ATR-FTIR spectra into refined data facilitated the construction of more resilient models [23,24]. Within this study, we explored various preprocessing methodologies, both independently and in combination, intending to ascertain the most suitable approach for our dataset. The techniques employed encompassed mean centering, autoscaling, smoothing (Savitzky–Golay), generalized least squares weighting (GLSW), and derivation (Savitzky–Golay) [23,24]. The selection of principal components was predicated upon the eigenvalue criterion, employing a graphical representation that correlates eigenvalues with the number of principal components. The identification of components took into consideration the highest cumulative explained variance. Lastly, a score graph was generated to scrutinize the differentiation between samples, thereby yielding enhanced insight into the chemical attributes and potential interactions within the formulations [23,24]. The PCA model was developed using MATLAB 7 and PLS-Toolbox 4 from the Eigenvector Research Group.

### 2.9. Spreadability

The parallel plate method was used to evaluate the spreadability profile and spreadability factor of the formulations [25]. A sufficient quantity of semisolid emulsions was meticulously placed into a central orifice with a diameter of 1 cm, positioned on a glass base marked with millimeter increments. Glass plates of known weights were then added sequentially to the sample at one-minute intervals. Following the addition of each plate, the average radius and diameter of the resulting spreading area were carefully measured, and the spreadability factor was calculated using Equation (2).
Sf (mm^2^/g) = A/W(2)
where Sf is the spreadability factor (mm^2^/g), A is the maximum spread area (mm^2^) after all plates were added, and W is the total weight added (g).

### 2.10. Dynamic Rheological Analysis

The rheological analysis was conducted to understand the semisolids’ flow, elasticity, and viscosity behavior. In rheology, fluids are classified based on their flow behavior under applied shear. Newtonian fluids exhibit a constant viscosity regardless of the shear rate; their flow behavior remains linear, meaning the viscosity does not change as the shear rate increases. In contrast, non-Newtonian fluids, like most semisolid emulsions, display a variable viscosity that depends on the shear rate. Among non-Newtonian types, pseudoplastic or shear-thinning fluids, as observed in this study, show a decrease in viscosity with increasing shear rate, which is advantageous for topical applications as it facilitates spreading on the skin.

Dynamic frequency sweep and thixotropic tests were performed using approximately 6 g of formulations in a Hybrid HR-10 rheometer (TA Instruments, New Castle, DE, USA). The rheometer utilized a cone plate geometry with a 40 mm diameter, 2° cone angle, and 0.3 mm gap. The dynamic oscillatory tests covered a frequency range of 0.1 to 600 rad/s with a constant stress of 1 × 10^−6^ MPa. The mechanical spectra recorded the dynamic moduli (G′ and G″) and η* as a function of frequency, where G′, G″, and η* represent the storage modulus (MPa), the loss modulus (MPa), and the dynamic viscosity (Pa.s), respectively. A flow ramp with a shear rate ranging from 100 to 0.1 s^−1^ was used for the thixotropic performance test. The sample interval was 1.0 s/pt, and each interval lasted for 60 s. All tests were conducted at 25 ± 2 °C.

### 2.11. Antioxidant Activity Evaluation

The assessment of antioxidant potential was conducted by measuring the scavenging capacity of the free radical 2,2-diphenyl-1-picrylhydrazyl (DPPH) [26]. Initially, a stock solution of DPPH at 5.05 mM in ethanol was prepared and stored in a freezer until testing. Before the tests, this solution was diluted to a final concentration of 50 µM DPPH by adding 50 µL of the stock solution to 4950 µL of ethanol. For the test, 1 g of each semisolid emulsion was weighed and mixed with 1.5 mL of absolute ethanol. The mixture was stirred for 5 min in a vortex, followed by 10 min of ultrasonic bath and subsequent centrifugation at 3000 rpm for 20 min. After centrifugation, 500 µL of the supernatant was combined with 500 µL of the DPPH solution (radical) or 500 µL of ethanol (control) and then incubated for 30 min at room temperature and protected from light. The absorbance of the samples was measured using a UV-VIS spectrophotometer (UV-1800 model, Shimadzu, Kyoto, Japan) at 517 nm. The antioxidant activity was calculated according to Equation (3):Scavenging Capacity (%) = 100 − [(AB_a_ − AB_b_)/AB_c_) × 100],(3)
where AB_a_ is the absorbance of the sample incubated with DPPH, AB_b_ is the absorbance of the blank, and AB_c_ is the absorbance of the negative control.

### 2.12. Occlusion Potential Determination

The in vitro water loss was determined as an indirect measure of occlusive potential [22]. To conduct the test, 50 mL of water was poured into a beaker, which was then covered with a layer of cellulose filter paper secured with an elastic band. Afterward, 200 mg of the formulations was evenly spread on the surface of the filter paper. A negative control beaker, prepared in the same way but without the formulation, was set up for comparison. A positive control was also established using solid petroleum jelly instead of the formulation. Following preparation, all the beakers were weighed using an analytical balance (AUW220D model; Marte Científica, Valinhos, Brazil) to determine their initial weight and then placed in an oven (37 °C; ECB-150D model; Biancodent equipamentos, Araucária, Brazil). After 24 h of incubation, the beakers were weighed again to calculate the amount of evaporated water, using Equation (4):Occlusion (%) = [(A − B)/A] × 100,(4)
where A is the water loss from the negative control beaker and B is the water loss from the sample beaker.

### 2.13. Bioadhesive Strenght

The bioadhesive strength of the semisolid formulations was tested using intact and injured swine ear skin as a biological membrane, provided by Frigorífico Padilha (Guarapuava, Brazil). The skin samples were sanitized and stored frozen at −20 °C until use. To simulate the injured skin condition, the samples were subjected to a superficial burn for 7 s on a heated plate (85 °C; HJ-3(ATRA) model, Ionlab, Araucária, Brazil) [24,27]. The skin was then cut into pieces approximately 4 cm in diameter and fixed to a glass plate. Then, 0.8 g of the formulation was placed on an upper support, positioned in contact with the biological membrane for 1 min, applying a force of 1 N. After this time, distilled water was added to a plastic tube on the opposite side until separation between the semisolid and the tissue was observed. The volume of water used until detachment was measured with a graduated cylinder to calculate the required quantity. The bioadhesive strength was determined based on Equation (5) [22,28].
Bioadhesive strength (dyne/cm^2^) = (V × G)/A,(5)
where V is the amount of water (g) required to detach the sample from tissue, G is the acceleration of gravity (980 cm/s^2^), and A is the area of exposed tissue (cm^2^).

### 2.14. Safety Assays

To evaluate the cytotoxicity of the semisolids, the fibroblast L-929 cell line (ATCC^®^ CCL-1™, Manassas, VI, USA) was cultured in DMEM supplemented with 10% fetal bovine serum (FBS), streptomycin (100 μg/mL), and penicillin (100 U/mL) at 37 °C in a 5% CO_2_ atmosphere (Cell incubator MCO-17 0ACL-PA model; PHC Corporation, Moriguchi, Japan). Initially, cells were seeded into 96-well plates at a concentration of 2.5 × 10^5^ cells/mL and subsequently exposed to varying concentrations of the samples, ranging from 1 to 1000 μg/mL. After 24 h of incubation, the 3(4-5-dimethyl)-2-5diphenyl tetrazolium bromide (MTT) salt reduction assay was performed. The fibroblasts were washed with PBS, and MTT (2 mg/mL) was added to each well. The cells were then incubated for 4 h at 37 °C. Following incubation, DMSO was introduced to dissolve the formazan crystals generated because of MTT reduction by mitochondrial enzymes. The absorbance was measured at 570 nm using a BIO-TEK Power Wave XS spectrophotometer (Hampton, NH, USA). MTT is initially a yellow-colored compound that transforms into a dark-blue hue upon metabolism by mitochondrial enzymes [29].

To evaluate the biocompatibility of the semisolids, a direct contact test was conducted based on Standard Practice for Assessment of Hemolytic Properties of Materials [30], using human erythrocytes. This study was previously approved by the Research Ethics Committee of the Federal University of Paraná-Brazil (#43948621.7.0000.0102). Peripheral blood of healthy human volunteers in tubes containing heparin was centrifuged (2000 rpm/5 min; Centribio 80-2B model; Biosystem, Campinas, Brazil), and the plasma fraction was removed. The resulting red blood cell was washed three times with 0.9% NaCl solution, and a 10% hematocrit suspension was prepared. Samples containing different weights of semisolids were placed into microtubes with 700 µL of 0.9% NaCl solution and left to equilibrate for about an hour. The tested concentrations correspond to 3.9, 7.81, 11.71, and 15.61 µg/mL of oil. Afterward, 100 μL of resuspended erythrocytes was added to the tubes. Positive and negative controls were prepared using distilled water or 0.9% NaCl solution, respectively. Additionally, blank samples were prepared with dispersed semisolids without erythrocyte suspension. Finally, the samples were incubated for 1 h at 37 °C, and the absorbance of the supernatant was measured in a spectrophotometer (540 nm; UV-1800 model, Shimadzu, Kyoto, Japan). The percentage of hemolysis was calculated according to the Equation (6) [24]:(6)$$\mathrm{Hemolysis}\;\left(\%\right)=\frac{\left(\mathrm{AbsA}-\mathrm{AbsB}\right)}{\mathrm{AbsC}}\times 100$$
where AbsA is the sample absorbance, AbsB is the blank absorbance, and AbC is the positive control absorbance.

### 2.15. Wound Healing Potential

For this assay, L-929 cells were plated at a 2.5 × 10^5^ cells/mL density in a 24-well plate and maintained in an incubator for 24 h. The following day, the cells were treated with DMEM supplemented with 0.5% FBS for 4 h. A wound was created in the cell monolayer at the center of the well using a sterile 20 µL pipette tip, followed by a careful PBS wash, and then treated with F2-KC-AO, F4-KC-BO, or F6-KC-B in supplemented DMEM (0.5% FBS) at a concentration of 7.81 µg/mL. Wound healing progress was observed at 0, 6, and 24 h time points using an inverted phase-contrast microscope (5× magnification; Olympus CKX41, Tokyo, Japan) [31].

### 2.16. Statistical Analysis

The results were obtained in triplicate and are expressed as mean ± standard deviation (SD). The statistical software GraphPad Prism^®^ version 8 was used for statistical analyses and figure creation. The normality of data distribution was checked using the Shapiro–Wilk test. The statistics were evaluated by one-way or two-way analysis of variance (ANOVA), followed by Tukey’s post-test. We defined significance levels as *p* < 0.05.

## 3. Results and Discussion

### 3.1. Polysaccharide-Based Semisolid Emulsion General Characterization

The investigation into topical pharmaceutical formulations has garnered significant attention due to their ability to enhance therapeutic efficacy while ensuring patient comfort during application [6,8,32]. In this context, it is important to examine GG and KG as stabilizing and gelling agents in vegetable oil emulsions for treating cutaneous wounds. These materials are prized for their multifunctional properties, biodegradability, and low toxicity. This makes them ideal for enhancing the safety and effectiveness of topical treatments, aligning with consumer preferences for natural and environmentally friendly products [8,33].

Vegetable oils have long been used in cutaneous products given their healing properties, especially in healing wounds. These materials contain essential fatty acids, antioxidants molecules, vitamins, and bioactive compounds that help skin regeneration and repair its protective barrier [12]. Remarkably, AO and BO exhibit anti-inflammatory, antioxidant, and antimicrobial properties, thereby contributing to the reduction in infection risk and acceleration of the healing process [12,34,35,36,37]. Their incorporation into semisolid formulations enhances the moisturization of injured skin, thus fostering an optimal environment for regenerating impaired tissues. After preparation, the formulations have a whitish color, homogeneous aspect, and shiny texture (Figure 2A), suggesting efficient oil incorporation into the gel matrix. The absence of phase separation or lumps further supports the initial stability of the emulsion, suggesting that the emulsification process effectively distributed the oil uniformly throughout the formulation.

Microscopic analysis (Figure 2B) shows oil droplets are dispersed within the biopolymer matrix (Figure 2B), which are crucial as the homogeneity and stability of semisolid emulsions are expected to ensure the final product’s effectiveness, particularly in topical applications. Regarding such aspects, we acknowledge the limitations of using 10× magnification for a detailed assessment of droplet sizes within the emulsions. The microscopy images presented were intended for a preliminary, qualitative evaluation of the formulation’s general organization and homogeneity. Future studies will focus on a more comprehensive characterization of the microstructure by employing advanced imaging techniques, such as confocal microscopy or scanning electron microscopy (SEM), along with droplet size analysis, to gain deeper insights into the stability and distribution of the emulsion components.

Reinforcing these findings, high physical stability was achieved after the centrifugation test. Appendix A shows no instability phenomena such as phase separation or cremation.

The physicochemical characterization of semisolid emulsions is summarized in Table 3. The pH values ranged from 4.74 to 5.06. Both F1-GG-AO and F3-GG-BO have slightly higher pH values (5.06 ± 0.07 and 4.97 ± 0.12, respectively; *p* < 0.05) compared to F2-KC-AO and F4-KC-BO (4.74 ± 0.11 and 4.74 ± 0.02, respectively). The pH values of these formulations fall within an acceptable range for topical application, ensuring compatibility with the skin’s natural pH and minimizing the risk of irritation [38]. These findings may be attributed to the different polysaccharides used, which could influence the acidity of the emulsions; KC tends to form formulations with a more acidic profile of pH values than GG due to the presence of sulfate ester groups in its molecular structure, which can dissociate in water, releasing hydrogen ions (H^+^) into the solution and lowering the pH values of the system [39,40]. Previous data showed that formulations become more acidic as the concentration of these dissociable sulfate groups increases [41]. In contrast, GG contains carboxylate groups that can release hydrogen ions as well, but typically to a lesser extent than the sulfate groups in KC [42]. Additionally, the degree of ionization and the buffering capacity of the GG may also play a role in maintaining a relatively higher pH than products prepared with KC. As a result, formulations with KC are generally more acidic than those with GG, reflecting the chemical nature of the polysaccharides [21].

The emulsions had similar densities, ranging from 1.0242 ± 0.0205 mg/mL to 1.0353 ± 0.0198 mg/mL (*p* > 0.05). This parameter suggests that the formulations have a pleasant consistency, which is essential for ensuring a uniform application and maintaining the structural integrity of the emulsion during storage and use [43].

The results of formulations without oils are described in the study of Alves and co-workers [21]. For pH, F5-GG-B presented values of 5.20 ± 0.22 and F6-KC-B of 4.65 ± 0.21, which are not statistically different from the formulations containing oils (*p* > 0.05). The same was observed for the density, in which values ranging from 0.99 ± 0.07 to 1.02 ± 0.03 for F5-GG-B and F6-KC-B were determined, respectively.

Considering our findings, it is important to highlight that no statistical tool was applied to rationally select the best composition for the formulation. The application of Design of Experiments (DoE) in formulation studies is invaluable, as it allows for a systematic evaluation of multiple variables and their interactions. This method yields more robust and consistent data on the effects of each component [44,45,46]. In this study, we opted not to use DoE, focusing instead on the feasibility of combining specific polysaccharides and vegetable oils, drawing from insights gleaned from prior research. We acknowledge this as a limitation, as incorporating DoE could have provided a deeper understanding of the optimization parameters for the formulation. For future studies, we recommend the implementation of DoE to further refine the most promising formulations identified here, ensuring a more optimized approach to the development of wound care products.

### 3.2. Infrared and PCA Model

Figure 3A depicts the raw material infrared spectra applied to prepare the formulations. These spectra exhibit characteristic peaks corresponding to the functional groups of each substance. Notably, bands within the 3000 to 3500 cm^−1^ region refer to the stretching vibrations of the O–H bonds, typical of alcohols and hydroxyl groups in polysaccharides and oils [47,48,49]. Additionally, bands within the 2850 to 2950 cm^−1^ range correspond to the asymmetric and symmetric C-H stretching prevalent in fatty acid chains, as observed in both oils [49,50]. In Figure 3B, the spectra represent the final formulations; they presented similar peaks within the regions around 2917 cm^−1^ and 2849 cm^−1^, signifying the presence of asymmetric and symmetric C–H stretching, respectively. Furthermore, the peaks at 1636 cm^−1^ may be linked to C=O or C=C stretching vibrations, characteristic of the bonds found in esters and unsaturated chains of oils [49]. The bands 1039 cm^−1^, 1061 cm^−1^, 1072 cm^−1^, and 1059 cm^−1^ could be associated with the C–O stretching vibration [48]. These spectra support the compatibility among the components. Lastly, the absence of new peaks suggests no chemical interaction altering the molecular structure of the excipients.

Previous studies indicated that GG and KC are chemically stable and demonstrate favorable interaction with plant actives, resulting in a gel matrix without compromising the structural integrity of the components [21]. The spectroscopic analysis supports this stability by revealing no significant alterations in the functional group peaks. Furthermore, the efficient integration of both oils suggests potential contributions to the bioactive properties of the formulation, as evidenced in previous research on natural products [49,50]. These findings underscore the promising potential of these formulations for developing topical delivery systems, given their capacity to uphold the chemical and functional integrity of the active ingredients.

Infrared spectroscopy, specifically FTIR spectroscopy, represents a potent technique utilized to identify and characterize functional groups of substances. Its application is prevalent in analyzing chemical compatibility between excipients and active ingredients within pharmaceutical formulations [51]. Despite its usefulness, the large and complex data produced by FTIR spectra, especially when analyzing component mixtures, requires the use of chemometric models. One such model, principal component analysis (PCA), helps simplify the analysis by identifying hidden patterns in the data and reducing its dimensionality. This makes it easier to interpret the results and develop more effective products. Combining infrared spectroscopy with a chemometric approach provides a deeper understanding of the chemical interactions between components, ultimately leading to better formulation optimization [23,24].

Figure 4A,B present the eigenvalues and the variance explained by each principal component. In Figure 4A, the first three principal components account for 87.24% of the total variance in the data. This indicates that these components capture most of the chemical information present in the raw materials, allowing for data analysis using only three principal components. Similarly, the formulations are also described by three principal components, which explain 87.16% of the total variance (Figure 4B). These findings suggest that the variance distribution in the formulations closely resembles that of the raw materials, with the first three components providing significant representation. In both cases, the preprocessing methods used were smoothing and GLSW.

The distribution of samples based on the principal components is represented in the three-dimensional score graphs (Figure 4C,D). In Figure 4C, depicting the raw material separation, PC1 explains 43.75% of the variance, PC2 explains 31.07%, and PC3 25.42%. This discrimination effectively distinguishes the various materials used in the formulations, such as AO, BO, GG, and KC. The polysaccharides were more proximate to the formulations than the oils, indicating a successful incorporation of the actives in the gel matrix. Concerning the formulations, PC1 explains 50.16% of the variance, PC2 explains 27.06%, and PC3 9.94% (Figure 4D). In this analysis, a distinct differentiation is observable between the formulations containing GG and KC, emphasizing these polysaccharides’ influence on the formulations’ ultimate chemical composition. These results illustrate that PCA effectively elucidated potential chemical interactions and similarities among the components.

### 3.3. Spreadability and Rheology Evaluations

The spreadability of semisolid formulations directly correlates with their capacity to cover a surface area, thereby indicating the effort required for proper application onto the skin [52]. The developed forms showed increased spreading area as more weight was applied, indicating the capacity for facilitated expansion when subjected to pressure (Figure 5A). This is an essential feature for cutaneous formulations, enabling the product to be distributed more easily over the skin, impacting dose uniformity and product acceptability [52,53]. In addition, the spreadability behavior is related to the rheological profile of semisolid formulations. In rheology, fluids are classified based on their flow behavior under applied shear. Newtonian fluids exhibit a constant viscosity regardless of the shear rate; their flow behavior remains linear, meaning the viscosity does not change as the shear rate increases. In contrast, non-Newtonian fluids exhibit a variable viscosity that depends on the shear rate [40,54]. Rheological measurements confirmed this behavior for the emulsions since all formulations’ complex viscosity (η*) decreased with increasing angular frequency, characteristic of pseudoplastic materials (Figure 5C).

The spreadability factor (Sf) of the formulations ranged from 7.13 to 8.47 mm^2^/g, signifying varying levels of application ease among them, with no statistically significant variances observed between them (Figure 5B, *p* > 0.05). These values are higher than those already described for other semisolid preparations for skin lesions, suggesting a more comfortable application in extensive and painful lesions, such as in some types of wounds [24,55,56]. Slightly higher Sf values were observed for formulations F2-KC-AO and F4-KC-BO, possibly due to the property of KC in forming gels with lower consistency and more flexible structure compared to GG, which tends to create more consistent structures [40,57]. The data on placebo formulations are presented by Alves and co-workers [21]. In this study, F5-GG-B had an Sf of 12.09 ± 0.33 mm^2^/g and F6-KC-B had 13.22 ± 1.26 mm^2^/g, which are higher than for our semisolids. This suggests that the oils increase the formulation consistency, as also observed by Alves and co-workers when chamomile oil was added to the GG and KC formulations [22].

The viscosity measurements corroborate this hypothesis, demonstrating that emulsions stabilized with GG present higher viscosity, reflecting a stronger network structure, especially in F3-GG-BO (Figure 5C). In contrast, KC forms a more fluid and elastic gel, ideal for application to painful wounds [58]. The lower viscosity of this polysaccharide is attributed to its sulfate ester groups, which create a less dense and more deformable semisolid structure [59]. Conversely, GG forms a more compact and structured gel network due to its carboxylate groups, which establish stronger intermolecular interactions, resulting in higher viscosity [60].

Figure 6 presents the rheological behavior of the semisolids. The graphs show the storage modulus (G′) and loss modulus (G″) as functions of the angular frequency (ω). For all samples, the G′ and G″ increases with angular frequency, indicating that the elastic and viscous behaviors intensify at higher frequencies. In all emulsions, G′ is consistently higher than G″ across the frequency range, suggesting a predominantly elastic behavior rather than viscous behavior. However, at around 600 rad.s^−1^, a shift occurs in the F2-KC-AO and F4-KC-BO formulations prepared with KC. In the F2-KC-AO formulation, G″ approaches G′, while in the F4-KC-BO formulation, G″ crosses G′, indicating that the gel becomes more fluid due to the increase in the loss modulus.

The findings align with the viscosity data, demonstrating that emulsions containing GG exhibit higher G′ values than those containing KC. This suggests a more robust and elastic network structure, with the F3-GG-BO emulsion displaying the highest G′ values, further emphasizing the strength of its gel structure.

Figure 7 shows a hysteresis loop in the stress (σ) versus shear rate (γ) curve for all semisolids, indicating thixotropic behavior. The area within the hysteresis loop suggests that the material’s structure is temporarily disrupted under shear, but it recovers gradually when the shear is removed, which is characteristic of thixotropic materials [61]. In Figure 7C, the formulation F3-GG-BO exhibits a more linear stress versus shear rate curve with a less pronounced hysteresis loop. This suggests a lower thixotropy degree, indicating that the emulsion’s structure is more stable under stress, and corroborating the spreadability factor, viscosity, G′, and G″ observations. The difference in thixotropy can be attributed to each polysaccharide’s distinct molecular structure and network interactions [62,63,64,65]. KC forms networks that are more prone to shear-induced breakdown but recover more slowly, resulting in a larger hysteresis area, as evidenced by the formulation of F4-KC-BO (Figure 7D). This slower recovery may be advantageous for applications on larger wounds where a longer time is required to cover the lesion adequately. This requires the formulation to remain more fluid for longer, allowing for a more comfortable application [66].

### 3.4. Antioxidant Activity

Figure 8 depicts the results of the antioxidant activity of semisolid emulsions. Depending on the composition, a distinct profile of scavenger action was observed. Both oils significantly improved the antioxidant potential for GG emulsions compared to placebo semisolid (F5-GG-B; *p* < 0.05). The incorporation of oils positively modulates the antioxidant capacity of GG [10,67]. This finding is particularly relevant when considering the formulation of topical applications for wound care, as higher antioxidant activity is beneficial for mitigating oxidative stress at the wound site, promoting better healing outcomes [6,24]. On the other hand, KC-based emulsions exhibited higher antioxidant properties than those stabilized with GG (*p* < 0.05). Remarkably, no statistical difference was observed when the oils were associated with the emulsions (*p* > 0.05). These data may indicate that KC exerts substantial antioxidant action, possibly due to its sulfated structure [68]. In wound healing, formulations based on KC may offer consistent antioxidant protection even without additional oils, making them potentially simpler and more cost-effective for therapeutic use. Different types of oils have varying effects on the antioxidant activity of emulsions, depending on the polysaccharide used. This highlights the importance of careful formulation design in products for treating wounds.

### 3.5. Occlusion Potential

The occlusion percentages were 35.76% ± 1.93%, 28.77% ± 0.16%, 21.27% ± 9.22%, and 28.55% ± 3.48% for F1-GG-AO, F2-KC-AO, F3-GG-BO, and F4-KC-BO, respectively (Figure 9). No significant differences were observed between the formulations (*p* > 0.05). However, F6-KC-B had a higher occlusion of 58.48% ± 11.46% compared to F5-GG-B (37.12% ± 1.28%) and other formulations based on *kappa*-carrageenan (F2-KC-AO and F4-KC-BO; *p* < 0.05). This result may be attributed to the hydrophilic properties of KC, which may form a gel network that attracts and retains water, acting as a humectant that may enhance skin hydration [68]. However, this gel network is sensitive to factors such as pH and the presence of hydrophobic substances [69], which may justify the negative impact of the oils in such properties, resulting in less cohesion and reduced water retention capacity.

Creating a moist environment is an important aspect of wound treatment, as it promotes healing by encouraging the migration of skin cells, the formation of new blood vessels, and the synthesis of connective tissue. However, the decision to use either occlusive or non-occlusive dressings should be based on the specific type of wound [58]. For formulations F1 to F4, which have occlusion percentages ranging from 21.27% to 35.76%, they are better suited for highly exudative wounds, such as deep wounds with a chronic inflammatory process, where the formulation needs to allow for controlled water loss [70]. In highly exuding wounds, it is crucial to control moisture to facilitate gas exchange and prevent microbial growth and maceration, which can hinder the healing process [58,71].

### 3.6. Bioadhesion

Bioadhesion is the process by which materials adhere effectively to the biological surface, prolonging their permanence in the site and reducing the frequency of reapplications, which can bring therapeutic benefits [6]. All formulations presented significantly higher bioadhesion in intact skin than in injured skin (Figure 10; *p* < 0.05). These findings support previous studies showing that topical formulations adhere more strongly to intact skin. This phenomenon can be explained by the hydrophobic interactions between the emulsion constituents and the lipids in the *stratum corneum* layer [72,73,74]. Thus, oils and hydrophobic groups in the GG and KC molecules may have contributed to this higher adhesion in intact skin. When cutaneous integrity is compromised, the loss of the *stratum corneum* exposes the viable epidermis, which is less hydrophobic, reducing the interactions responsible for emulsion adhesion [27]. This behavior is advantageous in wounds, as an overly adherent formulation can cause discomfort or pain during removal, especially in painful or sensitive wounds. Therefore, bioadhesion must be adjusted to ensure the permanence of the formulation without causing additional trauma to the scar tissue during removal [58,73].

Among the formulations, F3-GG-AO showed the highest bioadhesion on intact skin (~9000 Dyne/cm^2^; *p* < 0.05), in line with its rheological data that show higher viscosity and lower thixotropy, indicating a stronger and more resistant gel. Formulations F1-GG-AO and F2-KC-AO showed lower but similar bioadhesion (~6500 Dyne/cm^2^). At the same time, F4-KC-BO exhibited the lowest bioadhesive potential (~4500 Dyne/cm^2^, *p* < 0.05), consistent with their higher spreadability and lower viscosity values, suggesting a more fluid and less adherent structure.

In injured skin, F1-GG-AO and F3-GG-BO showed the highest bioadhesion values, which were similar to each other regardless of the type of vegetable oil used (5202 ± 180.20 and 4369 ± 540.60 Dyne/cm^2^, respectively; *p* > 0.05). The formulations with KC, F2-KC-AO, and F4-KC-BO showed lower bioadhesion, with superior results for avocado oil (3953 ± 180.20 and 1769 ± 360.4 Dyne/cm^2^, respectively, *p* < 0.05). Therefore, in injured skin, where minimizing pain and friction is essential, formulations with KC are promising to reduce painful stimuli and damage to remodeling tissue, facilitating the healing process [58,73].

### 3.7. Safety Assay

When developing topical products for wound healing, it is crucial to assess cellular toxicity, as a cytotoxic formulation could hinder tissue regeneration [24,75]. Figure 11 presents the results of the cytotoxicity of semisolids on the viability of L-929 cells by MTT assay. In all formulations tested, cell viability diminishes with increasing concentration.

To better understand the cytotoxicity, the CC_50_ was calculated (Table 4). Formulations based on KC presented higher values of CC_50_ in comparison to GG emulsions. These findings indicate that KC-based emulsions are less cytotoxic and safer than those prepared with GG, reinforcing its suitability for topical healing of wounds. Other studies have already indicated that KC has low cytotoxicity and promising cytoprotective potential [75,76]. The toxicity of GG may be due to the rigid three-dimensional structure of the semisolid emulsion, which could restrict the diffusion of nutrients and oxygen to fibroblasts, leading to greater cytotoxicity [77]. On the other hand, KC may form more flexible and less dense gels at similar concentrations, allowing for better exchange of nutrients and oxygen and resulting in less cytotoxicity. Additionally, KC is more water-soluble compared to GG, which tends to dissolve better at high temperatures. This could have resulted in low solubility of GG formulations in the cell medium. The higher solubility and dispersibility of KC may create a less hostile environment for fibroblasts than the denser and less soluble networks formed by GG, reflecting in the cell viability.

Based on the cytotoxicity findings, the biocompatibility of KC formulations was further assessed. Hemolysis is the destruction of red blood cell membranes, which releases hemoglobin and is a sensitive indicator of cellular toxicity [78]. The results indicated a hemolytic potential of less than 1% at all concentrations tested to the KC-based emulsions (Figure 12). This aligns with previous studies demonstrating that KC is non-hemolytic [79]. Furthermore, other research suggests that values below 5% indicate the biocompatibility of materials [80,81,82].

The hemolysis test is an essential assessment for determining the compatibility of formulations, especially when they may come into contact with blood or vascular tissues, such as topical formulations applied to wounds. When a formulation causes hemolysis, it indicates the potential for damage to blood cells, signaling poor biocompatibility. For products intended for topical use, especially on skin wounds, it is crucial that the formulation does not cause this adverse reaction [80,81]. Hemolytic products can be irritating and cause unwanted side effects when applied directly to the skin or to more sensitive areas, such as open wounds. Formulations that do not cause hemolysis, like those containing KC, are considered safe for topical use since they do not destroy red blood cells. This minimizes the risk of inflammatory reactions, does not compromise oxygen circulation, and promotes a healthier environment for cell regeneration [78,82,83].

### 3.8. Wound Healing

In Figure 13A, the images show the healing progress over time (0, 6, and 24 h) for the formulations tested. There is a continuous decrease in the open wound area over time, with the formulations containing oils demonstrating a greater degree of cell migration, indicating a healing effect. Figure 13B illustrates the percentage of open wound area at different time points (0, 6, and 24 h) for the specified formulations compared to the negative control. The findings suggest that the F4-KC-BO and F2-KC-AO formulations have a positive impact on healing, demonstrating a significant reduction in the open wound area when compared to both the negative control and the F6-KC-B formulation (without oil), mainly after 6 h. The F4-KC-BO formulation (red) exhibits a faster healing rate, followed by the F2-KC-AO formulation (green), while the F6-KC-B formulation displays lower efficacy in promoting healing due to its moisturizing action and the fatty acids that promote cell regeneration.

BO stands out for its high content of essential fatty acids, such as gamma-linolenic acid and alpha-linolenic acid, which play crucial roles in maintaining the integrity of cell membranes and supporting skin regeneration. Studies suggest that these fatty acids accelerate the healing process by promoting wound re-epithelialization and reducing inflammation, which is crucial for tissue recovery. In addition, BO is rich in tocopherols and carotenoids, compounds with potent antioxidant properties that reduce oxidative stress at the wound site, further promoting tissue repair [84,85]. Similarly, AO contains high levels of oleic and palmitic acid, known for its moisturizing and anti-inflammatory properties. Oleic acid improves the penetration of active compounds across the skin barrier, aiding in the delivery of essential nutrients to the wound site. Additionally, AO is rich in vitamins A, D, and E, aiding collagen synthesis and providing antioxidant protection. This combination of moisturizing, regenerative, and antioxidant effects is particularly beneficial for wound healing [86,87]. In summary, adding BO and AO to *kappa*-carrageenan-based formulations significantly improved in vitro wound healing. These findings provide valuable insights for future studies on wound healing, suggesting that the incorporation of blackcurrant and avocado oils could be explored further for their potential to enhance tissue regeneration and expedite wound closure.

Importantly, the evaluation of skin formulation safety and efficacy is most effective when conducted using models that closely replicate human skin physiology. Such models provide essential insights into complex immune responses, cell interactions, and tissue architecture—elements that simpler cell line models fail to capture. In this study, we utilized in vitro cell line models to gather preliminary data on biocompatibility and wound healing potential. However, we recognize that this limits our ability to directly translate these findings to human skin. Future research will focus on employing more advanced models, such as ex vivo human skin or animal models, to achieve a more precise understanding of the safety and therapeutic potential of these formulations in contexts that more accurately reflect the wound healing process in humans.

## 4. Conclusions

This study successfully developed and evaluated GG and KC-based semi-solid emulsions containing AO or BA for wound healing applications. The formulations exhibited suitable physicochemical characteristics for topical use, such as appropriate pH and compatibility between excipients, ensuring formulation stability and safety. Furthermore, rheological properties, including pseudoplastic behavior and thixotropy, combined with favorable spreadability results, contribute to ease of application. Lower bioadhesion on injured tissues increases patient comfort, reducing the risk of trauma during removal.

The KC formulations exhibited superior antioxidant activity and safety. Moreover, the in vitro wound healing assays revealed that adding AO and BO significantly increased cell migration, indicating their potential to accelerate wound closure. These results suggest that the developed emulsions, particularly those based on *kappa*-carrageenan, offer promising potential as safe, effective, and comfortable topical formulations for wound care. Future studies should focus on further optimizing these formulations and evaluating their performance in more complex in vivo wound models.

## Figures and Tables

**Figure 1 pharmaceutics-16-01426-f001:**
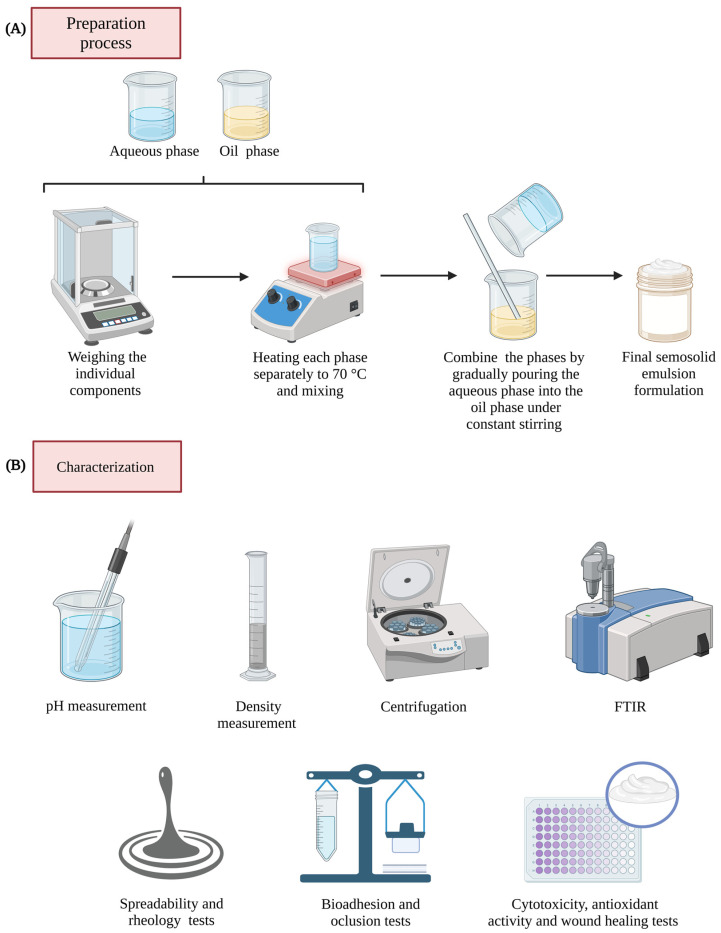
Flowchart of the formulation and characterization procedures. The preparation of the semisolid emulsion involves several sequential steps (**A**): weighing the individual components for the oil phase (OP) and aqueous phase (AP), heating each phase separately to 70 °C to ensure proper dissolution and mixing, combining the phases by gradually pouring the aqueous phase (AP) into the oil phase (OP) under constant stirring to form a uniform emulsion, and obtaining the final gel–cream formulation. The emulsion was subsequently characterized through various analyses (**B**): Fourier-transform infrared spectroscopy (FTIR) to assess molecular interactions and confirm compatibility among components, centrifugation to evaluate physical stability and detect any phase separation, spreadability and reology testing to determine ease of application and coverage on the skin, density measurement to assess formulation consistency, pH measurement with a pH meter to ensure suitability for skin application, bioadhesion and occlusion potential, antioxidant activity, cytotoxicity testing using cell cultures to evaluate biocompatibility and potential safety for skin use, and wound healing assay to determine efficacy.

**Figure 2 pharmaceutics-16-01426-f002:**
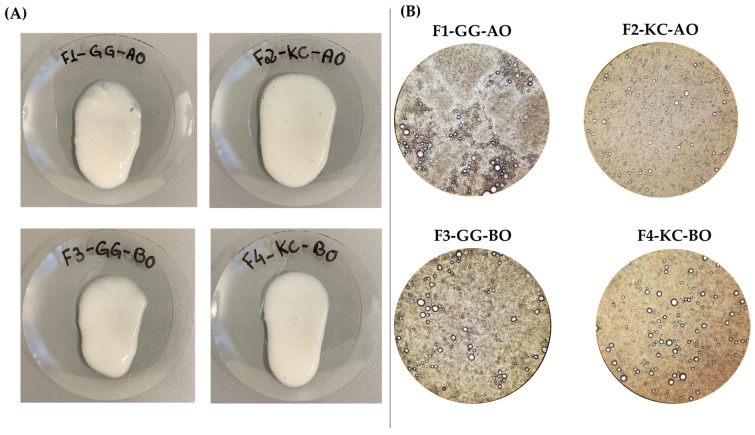
Macroscopic (**A**) and microscopic (**B**) images of polysaccharide-based semisolid emulsions containing vegetable oils. Overall, the formulations have a whitish color, homogeneous aspect, and shiny texture. The microscopic evaluation indicates that the system effectively dispersed the oil, keeping it stable within the semisolid structure. Abbreviations: GG—Gellan gum; KC—*Kappa*-carrageenan; BO—Blackcurrant Oil; AO—Avocado Oil.

**Figure 3 pharmaceutics-16-01426-f003:**
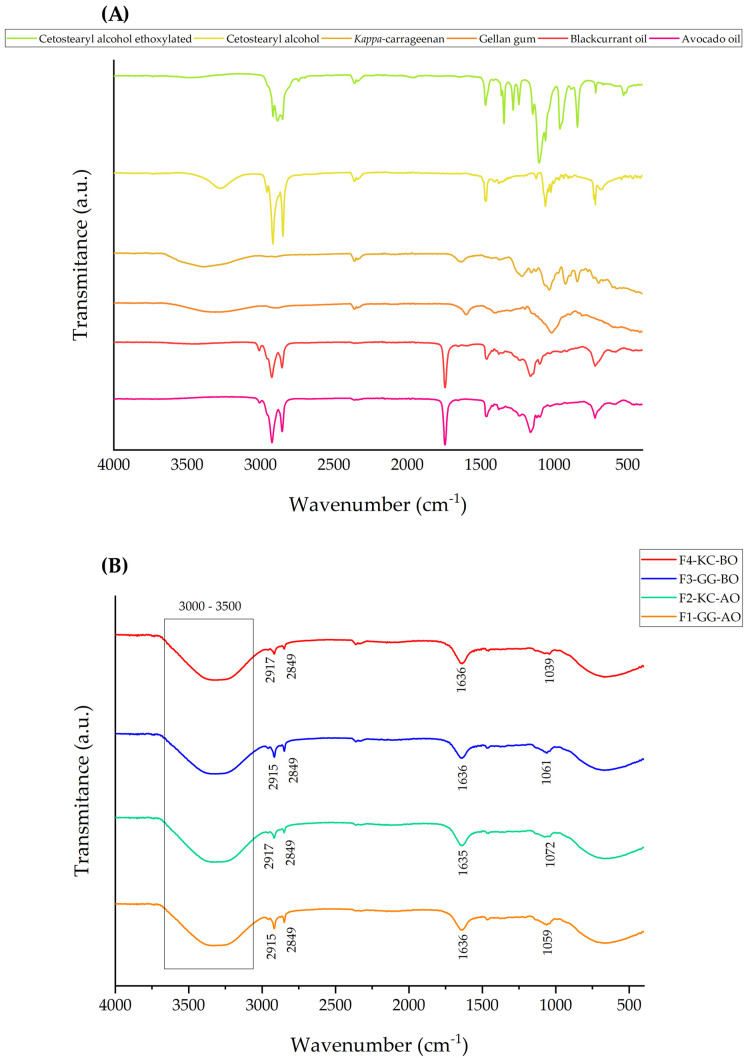
Infrared spectra of raw materials (**A**) and semisolid emulsions (**B**). The spectra exhibit characteristic peaks corresponding to the functional groups present in the substances. Additionally, these spectra support the compatibility among the components, as the absence of significant new peaks suggests no chemical interaction altering the molecular structure of the excipients.

**Figure 4 pharmaceutics-16-01426-f004:**
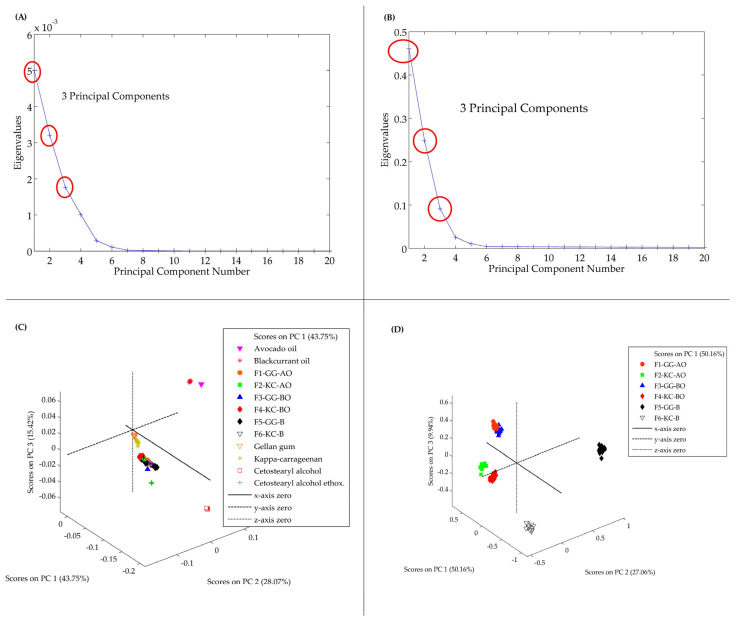
PCA model. In (**A**,**B**) are the eigenvalues graphs, which indicate that these three principal components encompass most of the chemical information in the raw materials. The red circles represent the principal components selected for the model. In (**C**,**D**) are the score plot graphs that reveal a distinct differentiation is observable between the formulations containing GG and KC, emphasizing these polysaccharides’ influence on the formulations’ ultimate chemical composition.

**Figure 5 pharmaceutics-16-01426-f005:**
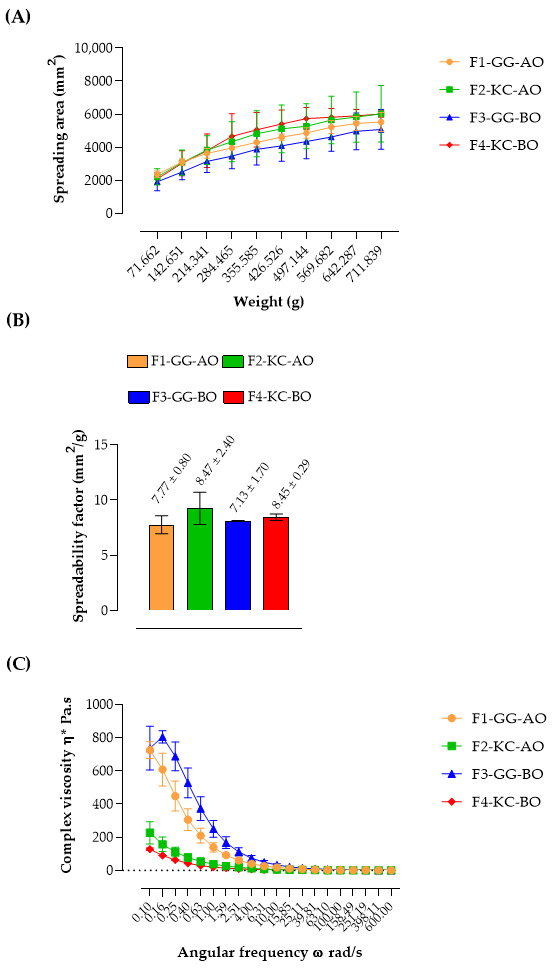
Spreadability profile (**A**), spreadability factor (**B**), and viscosity (**C**) of semisolid emulsions. The developed emulsions demonstrated an increased spreading area with the application of more weight, suggesting they can expand more easily under pressure. Moreover, rheological measurements supported this behavior, as the complex viscosity (η*) of all formulations decreased with increasing angular frequency, which is a characteristic of pseudoplastic materials.

**Figure 6 pharmaceutics-16-01426-f006:**
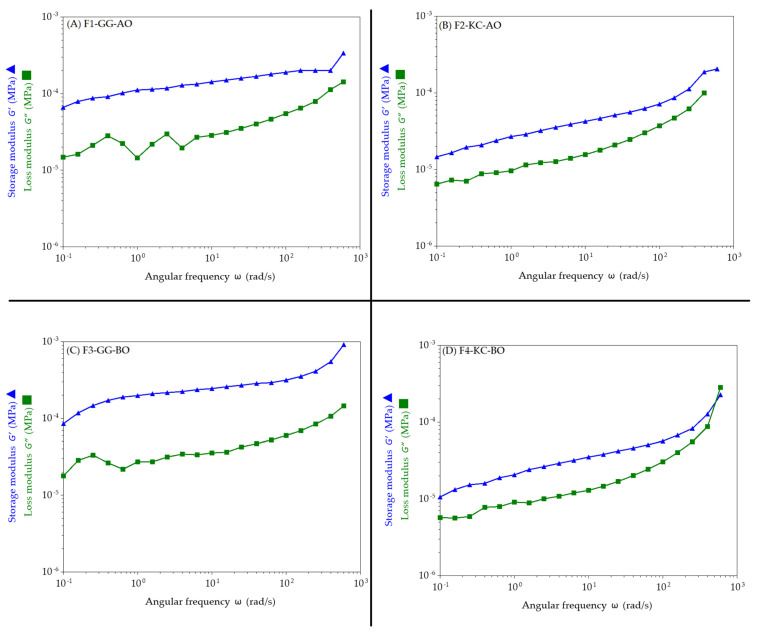
Storage modulus (G′) and loss modulus (G″) as functions of angular frequency (ω). In (**A**,**B**) formulations containing AO stabilized with GG and KC, respectively. In (**C**,**D**) formulations prepared with BO stabilized with GG and KC, respectively. Data indicates that elastic and viscous behaviors become more pronounced at higher frequencies, suggesting a predominantly elastic rather than viscous behavior.

**Figure 7 pharmaceutics-16-01426-f007:**
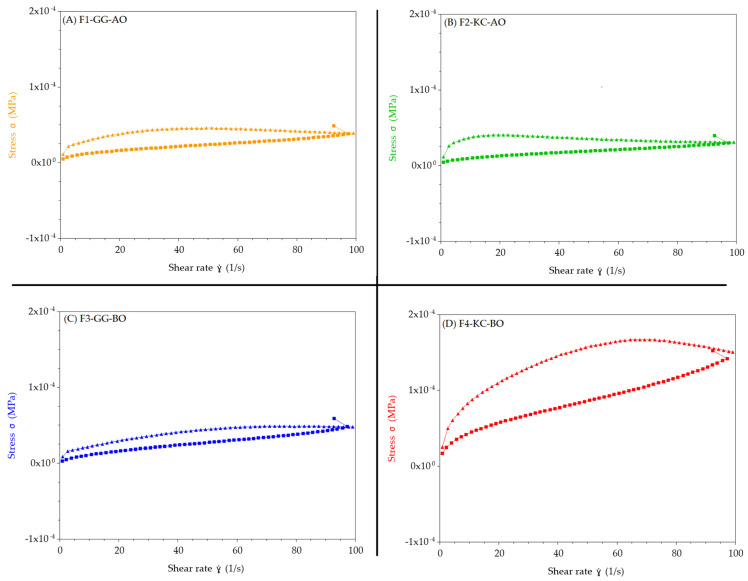
Thixotropy evaluation of F1-GG-AO (**A**), F2-KC-AO (**B**), F3-GG-BO (**C**), and (**D**) F4-KC-BO. The data show that the material’s structure is temporarily disrupted under shear, but it recovers gradually when the shear is removed, which is characteristic of thixotropic materials.

**Figure 8 pharmaceutics-16-01426-f008:**
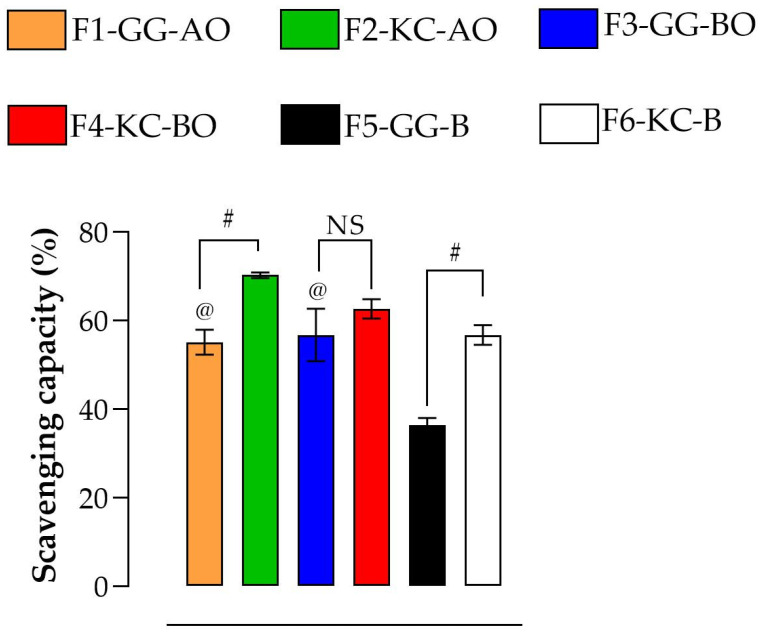
Antioxidant activity. The @ denotes the significant difference (*p* < 0.05) between formulations and their respective blank forms (F1-GG-AO versusF5-GG-B, and F3-GG-BO versus F5-GG-B); # represents the significant difference (*p* < 0.05) between polysaccharides (F1-GG-AO versus F2-KC-AO, and F5-GG-B versus F6-KC-B). NS means “not significant”. Both oils significantly enhanced the antioxidant potential of GG emulsions compared to the placebo semisolid, while emulsions stabilized with KC demonstrated higher antioxidant properties than those stabilized with GG.

**Figure 9 pharmaceutics-16-01426-f009:**
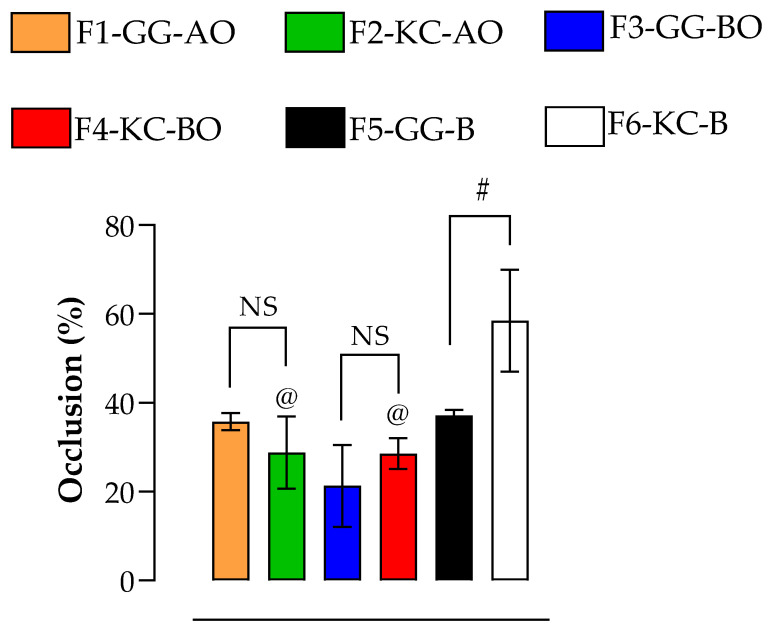
Occlusion potential. The @ denotes the significant difference (*p* < 0.05) between formulations and their respective blank forms (F2-KC-AO versusF6-KC-B, and F4-KC-BO versus F6-KC-B); # represents the significant difference (*p* < 0.05) between polysaccharides (F5-GG-B versus F6-KC-B). NS means “not significant”. Similar occlusion potential was observed among the formulations. Data also suggests that the oily components may negatively affect the KC formulations.

**Figure 10 pharmaceutics-16-01426-f010:**
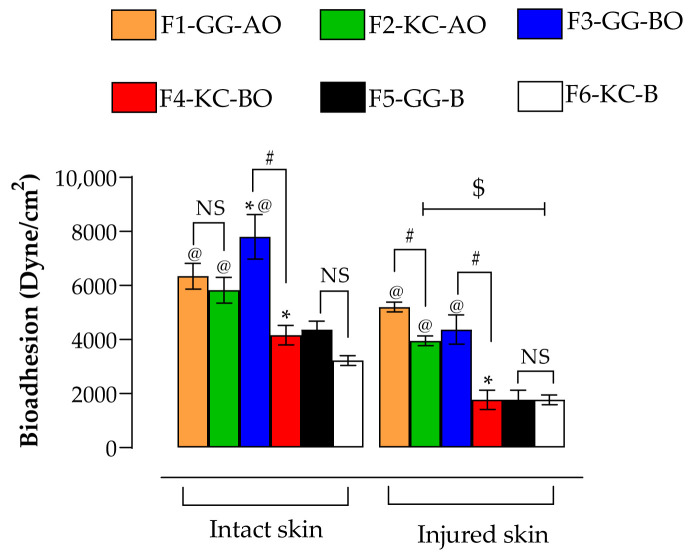
Bioadhesion potential in intact and injured skin. The @ denotes the significant difference (*p* < 0.05) between formulations and their respective blank forms; # represents the significant difference (*p* < 0.05) between polysaccharides with the same oil (F1-GG-AO versusF2-KC-AO, or F3-GG-BO versus F4-KC-BO); * denotes the significant difference (*p* < 0.05) between oils with the same polysaccharide (F1-GG-AO versus F3-GG-BO or F2-KC-AO versus F4-KC-BO); and $ represents the significant difference (*p* < 0.05) between intact and injured skin. NS means “not significant”. All formulations presented significantly higher bioadhesion in intact skin than in injured skin.

**Figure 11 pharmaceutics-16-01426-f011:**
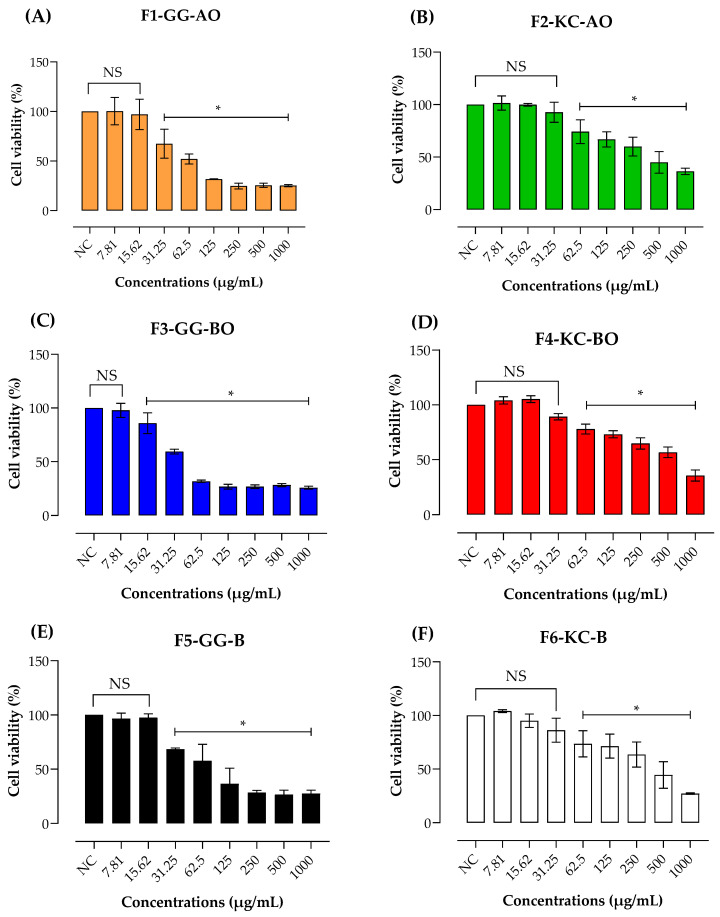
Effect of F1-GG-AO (**A**), F2-KC-AO (**B**), F3-GG-BO (**C**), F4-KC-BO (**D**), F5-GG-B (**E**), and F6-KC-B (**F**) (1–1000 µg/mL) on the viability of L-929 cells by MTT assay. A negative control (non–treated cells) was conducted and considered 100% viability. Mean values were calculated from 3 independent results. The * denotes the significative difference from the negative control (*p* < 0.05). NS means “not significant”. In all formulations examined, the viability of cells is observed to decline as the concentration increases.

**Figure 12 pharmaceutics-16-01426-f012:**
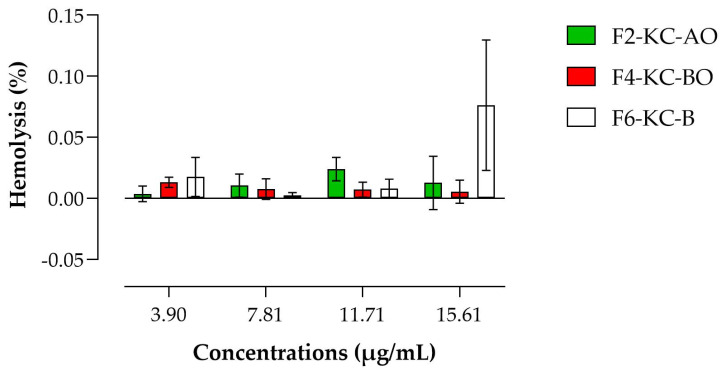
Hemolytic assay of KC semisolid emulsions. The results showed a hemolytic potential of less than 1% for all tested concentrations of the KC-based emulsions.

**Figure 13 pharmaceutics-16-01426-f013:**
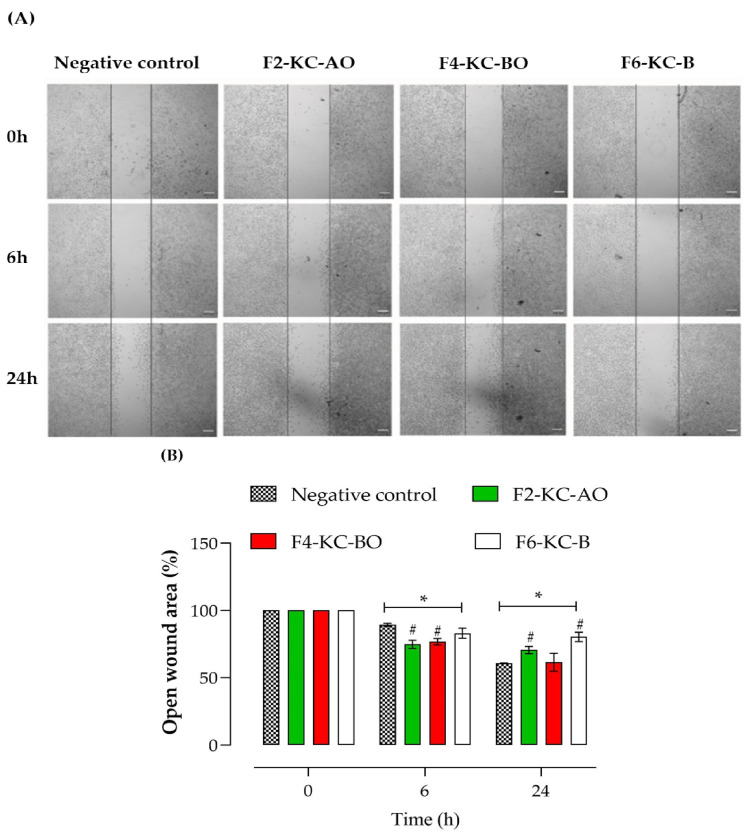
Representative images showing the progression of healing over time (**A**) and percentage of open wound area at different times (0, 6, and 24 h) (**B**) for the F2-KC-AO, F4-KC-BO, and F6-KC-B, compared to the negative control. The * denotes the significant difference (*p* < 0.05) with time zero in the same group, and # denotes the significant difference (*p* < 0.05) with negative control at the same time. There is a consistent reduction in the area of open wounds over time, with formulations containing oils exhibiting a more pronounced degree of cell migration, which suggests an effective healing process.

**Table 1 pharmaceutics-16-01426-t001:** Quali-quantitative composition of formulations.

Phase	Constituents	Concentration (%, *w*/*w*)
Oily phase	Cetostearyl Alcohol Ethoxylate	3.0
	Cetostearyl Alcohol	5.0
	Oil (AO or BO)	3.0
	Vitamin E	0.2
Aqueous phase	Polysaccharide (GG or KC)	0.5
	EDTA	0.1
	Propylene glycol	5.0
	Nipaguard^®^	0.3
	Water^sqf^	100

Abbreviations: EDTA—Ethylenediamine tetraacetic acid; GG—Gellan gum; KC—*Kappa*-carrageenan; BO—Blackcurrant Oil; AO—Avocado Oil.

**Table 2 pharmaceutics-16-01426-t002:** Names and qualitative composition of formulations.

Formulations	Gellan Gum (GG)	*Kappa*-Carrageenan (KC)	Avocado Oil (AO)	Blackcurrant Oil (BO)
F1-GG-AO	X	-	X	-
F2-KC-AO	-	X	X	-
F3-GG-BO	X	-	-	X
F4-KC-BO	-	X	-	X
F5-GG-B	X	-	-	-
F6-KC-B	-	X	-	-

Abbreviations: GG—Gellan gum; KC—*Kappa*-carrageenan; BO—Blackcurrant Oil; AO—Avocado Oil. The “X” indicates the presence of the component in the formulation.

**Table 3 pharmaceutics-16-01426-t003:** Physicochemical characterization of semisolid emulsions.

Formulations	pH	Density (mg/mL)
F1-GG-AO	5.06 ± 0.07 ^#^	1.0315 ± 0.0325
F2-KC-AO	4.74 ± 0.11	1.0242 ± 0.0205
F3-GG-BO	4.97 ± 0.12 ^#^	1.0294 ± 0.0259
F4-KC-BO	4.74 ± 0.02	1.0353 ± 0.0198

^#^ difference between polysaccharides (F1 versus F2 or F3 versus F4).

**Table 4 pharmaceutics-16-01426-t004:** CC_50_ of semisolid emulsions against L-929 cells.

Formulation	CC_50_ (µg/mL)
F1-GG-AO	63.13 ± 4.35
F2-KC-AO	390.03 ± 8.06 ^#^
F3-GG-BO	40.77 ± 1.79
F4-KC-BO	589.69 ± 85.47 ^#^*
F5-GG-B	65.79 ± 14.25
F6-KC-B	541.24 ± 9.16 ^#^

CC_50_: Cytotoxic concentration for 50% of cells. The * denotes the significant difference (*p* < 0.05) between oils with the same polysaccharide (F1-GG-AO versus F3-GG-BO or F2-KC-AO versus F4-KC-BO); ^#^ represents the significant difference (*p* < 0.05) between polysaccharides with the same oil (F1-GG-AO versus F2-KC-AO, or F3-GG-BO versus F4-KC-BO).

## Data Availability

The original contributions presented in the study are included in the article/Appendix A, further inquiries can be directed to the corresponding author/s.

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
