# Peer review of "Polysaccharide-Stabilized Semisolid Emulsion with Vegetable Oils for Skin Wound Healing: Impact of Composition on Physicochemical and Biological Properties"

_pharmaceutics, 2024, doi:10.3390/pharmaceutics16111426_

Round 1

Reviewer 1 Report

Comments and Suggestions for Authors

I have read with interest the manuscript of Morais Trindade et al on a semisolid emulsion with vegetable oils for explore its potential application in promoting for skin wound healing. The characterization of the formulations is based on physical stability, rheological properties and bioadhesion. In addition, biocompatibility was estimated through cytotoxicity assays. Finally authors examined the healing potential of the formulations developed in an in vitro wound model. My suggestion is to publish this work after minor revisions, see below.

1.     The authors should describe more correctly the equipment used. Please add company, city and country. For example pH-meter, microscope,centrifugation test, FTIR spectrometer,….

2.     The preparation process of the developed formulations is confusing. When are the selected oils (AO, BO) added? Before or after emulsifying? If it is later, why? Can't they be added during the emulsification process? Why? reasons? Explain it

3.     Describe in more detail the data during elaboration process: Speed, equipment used, time, temperature,...

4.     Describe the function of cetostearyl alcohol ethoxylate and cetosterayl alcohol ingredients. What type of emulsifier? Add HLB.

5.     How and when the KC and GC components are added in the preparation process of emulsions?. Explain it

6.     Explain the preparation technique for microscope visualization.

7.     Is 10X magnification sufficient to correctly observe the oil droplets? Explain it

8.     The microscope images must have the tool to measure the drops size. Please add.

9.     Did the authors perform large-term stability studies about formulations developed?

10.    Do the authors perform flow curves in the rheological analyses? Please describe and introduce flow types (Newtonian or non-Newtonian) to improve rheological section.

11.  Change minutes for min

12.  I suggest to the authors introduce the word emulsion in keywords section

13.  Where is chapter 4?

Author Response

REVISOR 1

I have read with interest the manuscript of Morais Trindade et al on a semisolid emulsion with vegetable oils for explore its potential application in promoting for skin wound healing. The characterization of the formulations is based on physical stability, rheological properties and bioadhesion. In addition, biocompatibility was estimated through cytotoxicity assays. Finally authors examined the healing potential of the formulations developed in an in vitro wound model. My suggestion is to publish this work after minor revisions, see below.

Answer: We would like to express our gratitude to the reviewer for the time and attention dedicated to reading our article and for considering its publication. We have carefully addressed and responded to your questions and comments, and we have highlighted the revisions made to the text in green.

  1. The authors should describe more correctly the equipment used. Please add company, city and country. For example pH-meter, microscope, centrifugation test, FTIR spectrometer,….

Answer: Thank you for the suggestion. We agree that including this information makes the article more complete and easier to understand. The correction has been made, and the data has been added to the revised manuscript, which is highlighted in green.

  1. The preparation process of the developed formulations is confusing. When are the selected oils (AO, BO) added? Before or after emulsifying? If it is later, why? Can't they be added during the emulsification process? Why? reasons? Explain it

Answer: Thank you for your insightful question. The selected oils, avocado oil (AO) and blackcurrant oil (BO), were added after the emulsification process. This choice was made to preserve the bioactive properties of these oils, which could be compromised if exposed to high temperatures during emulsification. During the preparation, the emulsification process involves heating the aqueous and oily phases separately, as detailed in the methodology. If AO and BO were added during this stage, the elevated temperatures (around 70°C - 75°C) might lead to the degradation of thermolabile compounds present in these oils, such as unsaturated fatty acids, vitamins, and antioxidants. By adding the oils after the emulsification process, when the formulation has cooled to room temperature, we aim to maintain the integrity and efficacy of these bioactive compounds, ensuring that the final product retains its desired therapeutic properties. Therefore, this method not only allows the emulsification to occur effectively but also maximizes the preservation of the functional characteristics of the oils, which are crucial for the wound healing activity of the formulations.

            For enhanced comprehension, a brief explanation was included in the revised document, which is highlighted in green (Lines 122 to 142).

  1. Describe in more detail the data during elaboration process: Speed, equipment used, time, temperature,...

Answer: Thank you for your request for more detailed information on the formulation process. Below is a comprehensive description of the parameters used:

Speed and Equipment: The emulsification was performed manually, using a glass rod to ensure constant stirring at a low speed. This method was chosen to allow for a gentle mixing process, which helps to form a stable emulsion without the risk of overheating or over-shearing the components.

Time: The manual stirring was maintained for approximately 15 minutes to ensure proper integration of the aqueous and oily phases, resulting in a smooth and homogenous texture.

Temperature: During the preparation, the components of the oily phase were heated to 70°C until fully melted, while the aqueous phase was heated to 75°C. Once both phases reached the desired temperatures, the aqueous phase was gradually added to the oily phase with continuous manual stirring. The mixture was allowed to cool down to room temperature (25°C) before adding the avocado oil (AO) or blackcurrant oil (BO), along with Nipaguard®. Adding these ingredients at lower temperatures helps preserve the bioactive properties of the oils, which may degrade if exposed to high heat.

Storage Conditions: After preparation, the final emulsions were stored in plastic containers at room temperature to maintain their stability until further analysis.

These parameters were carefully selected to optimize the stability and quality of the emulsions while preserving the bioactive properties of the oils. For enhanced comprehension, a brief explanation was included in the revised document, which is highlighted in green (Lines 122 to 142).

  1. Describe the function of cetostearyl alcohol ethoxylate and cetosterayl alcohol ingredients. What type of emulsifier? Add HLB.

Answer: Thank you for your insightful comment regarding the role of cetostearyl alcohol ethoxylate and cetostearyl alcohol in our formulations. Cetostearyl alcohol ethoxylate functions as a non-ionic oil-in-water (O/W) emulsifier, reducing interfacial tension between the aqueous and oily phases to create a stable emulsion, with an HLB value typically ranging between 12-15. Cetostearyl alcohol, a mixture of fatty alcohols (cetyl and stearyl alcohols), acts as a co-emulsifier and thickening agent, enhancing the formulation's viscosity and texture. With a lower HLB value of around 5-6, its lipophilic nature complements the primary emulsifier, providing additional stability. Together, these components ensure a balanced emulsification system that enhances the formulation's uniformity, spreadability, and effectiveness for topical application, contributing to the product's suitability for wound healing. For enhanced comprehension, a brief explanation was included in the revised document, which is highlighted in green (Section 2.2).

  1. How and when the KC and GC components are added in the preparation process of emulsions?. Explain it

Answer: Thank you for your insightful comment. The kappa-carrageenan (KC) and gellan gum (GG) were added during the preparation of the aqueous phase of the emulsions. Specifically, KC or GG was dissolved in the heated aqueous phase at 75°C to ensure complete dissolution and hydration. This step is crucial because both KC and GG need sufficient heat and mixing to form a gel-like matrix, which enhances the stability and viscosity of the final emulsion. Once the KC or GG was fully dissolved, the aqueous phase was combined with the oily phase while continually stirring by hand to create a uniform emulsion. This process ensures that the polysaccharides are well-dispersed, providing structural support and improving the rheological properties of the formulations. For enhanced comprehension, a brief explanation was included in the revised document, which is highlighted in green (Section 2.2).

  1. Explain the preparation technique for microscope visualization.

Answer: Thank you for your insightful comment. For microscopic analysis, a small aliquot of the emulsion was carefully placed on a clean glass slide. A coverslip was gently positioned over the sample to create a uniform, thin layer, which facilitates optimal visualization under the microscope. The prepared slide was then examined under an optical microscope at various magnifications, allowing for detailed observation of the emulsion's microstructure. This technique was used to assess the homogeneity, droplet size distribution, and dispersion of the oil phase within the continuous phase, providing valuable insights into the stability, consistency, and effectiveness of the emulsifying system used in the formulations. For enhanced comprehension, a brief explanation was included in the revised document, which is highlighted in green (Section 2.3).

  1. Is 10X magnification sufficient to correctly observe the oil droplets? Explain it

Answer: Thank you for your insightful comment regarding the use of 10X magnification for observing the oil droplets. We acknowledge that while 10X magnification provides a basic view of the emulsion's microstructure, it has certain limitations in terms of resolution, particularly for detailed observation of smaller droplets. The primary objective of this technique was to perform a qualitative assessment of the preliminary organization and homogeneity of the formulations rather than a detailed quantitative characterization. We recognize this as a limitation of our current study and will include a brief statement in the discussion section to acknowledge these constraints. We agree that additional, higher-resolution imaging techniques, such as using 40X or 100X magnification, or even electron microscopy, would be necessary to obtain a more comprehensive understanding of the differences in the structural organization of the formulations. These further assessments could provide more precise insights into the droplet size distribution and stability of the emulsions.

For enhanced comprehension, a brief explanation was included in the revised document, which is highlighted in green (Lines 370 to 377).

  1. The microscope images must have the tool to measure the drops size. Please add.

Answer: Thank you for your valuable suggestion regarding the inclusion of a tool to measure droplet size in microscope images. We understand the importance of precise droplet size measurement for a detailed characterization of emulsions. However, the primary purpose of the microscopy images in this study was to provide a qualitative visualization of the formulation's organization and general homogeneity, rather than a detailed, quantitative analysis of droplet sizes.

Our approach was intended to offer a preliminary view of how the components are dispersed within the emulsion, aligning with the qualitative nature of our assessment. We acknowledge that a more comprehensive analysis, including precise measurements of droplet sizes, could provide further insights into the stability and performance of the formulations. We will include a statement in the discussion to recognize this limitation and emphasize that future studies will aim to explore these aspects in greater depth. In particular, we plan to employ more advanced imaging techniques, such as confocal microscopy or scanning electron microscopy (SEM), combined with droplets size analysis (e.g., dynamic light scattering or laser diffractometry), to obtain a more detailed understanding of the microstructural properties of the emulsions.

For enhanced comprehension, a brief explanation was included in the revised document, which is highlighted in green (Lines 370 to 377).

  1. Did the authors perform large-term stability studies about formulations developed?

Answer: Thank you for your question about the long-term stability studies of the developed formulations. We agree that conducting these studies would be valuable for understanding the shelf life and performance of the emulsions. However, it is important to note that this preliminary study focused primarily on the formulations' initial development and characterization to evaluate their potential for wound healing. As this research aims to establish a foundation for further development, we did not conduct extensive long-term stability studies. Future investigations, particularly with an optimized formulation version, are planned to include comprehensive stability assessments under various storage conditions. These studies will help evaluate key parameters such as phase separation, consistency, and bioactive compound retention over prolonged periods, ensuring the efficacy and reliability of the formulations for clinical use.

  1. Do the authors perform flow curves in the rheological analyses? Please describe and introduce flow types (Newtonian or non-Newtonian) to improve rheological section.

Answer: Thank you for your comment. In this study, we performed dynamic frequency sweep and thixotropic tests to evaluate the formulations' flow, elasticity, and viscosity. The rheological data were obtained using a Discovery HR10 rheometer with cone plate geometry, focusing on capturing the mechanical spectra (G′, G″, and η*) across a frequency range of 0.1 to 600 rad/s. We also performed a thixotropic test to assess structural recovery at varying shear rates.

However, we acknowledge that an explicit description of the flow behavior (Newtonian vs. non-Newtonian) was not included in our initial discussion. Based on the flow ramp results, we have identified that our formulations exhibit non-Newtonian, pseudoplastic behavior, as demonstrated by the shear-thinning profile and recovery observed in the thixotropic test. This finding aligns with the behavior expected in semisolid emulsions, where viscosity decreases with increasing shear rate, an advantageous feature for topical applications.

We incorporated this clarification in the revised manuscript, adding details on the flow type observed and its relevance to the formulations' application. We appreciate this suggestion, which has strengthened our discussion of the rheological properties.

            For enhanced comprehension, we included in the revised document additional information about the topic, which is highlighted in green (Lines 231 to 238).

  1. Change minutes for min

Answer: We apologize for the mistake and appreciate the correction. The adjustment has been made and highlighted in green in the revised manuscript.

  1. I suggest to the authors introduce the word emulsion in keywords section

Answer: We appreciate the suggestion, as it will indeed facilitate the search and discovery of our work. The term has been added.

  1. Where is chapter 4?

Answer: Thank you for bringing this to our attention. We acknowledge that there was an error in the numbering of the sections within the manuscript. After section 3 (Results and Discussion), the numbering inadvertently skipped to section 5 (Conclusion), which created the appearance of a missing chapter. We apologize for any confusion this may have caused. We have corrected the section numbering in the revised manuscript to ensure proper order and clarity. We would like to assure you that no content or chapters were omitted, and all intended sections are present in the document.

Reviewer 2 Report

Comments and Suggestions for Authors

Polysaccharide-Stabilized Semisolid Emulsion with Vegetable Oils for Skin Wound Healing: Impact of Composition on Physicochemical and Biological Properties.

The overall goal of this study was to utilize avocado (Persea Gratissima) and blackcurrant (Ribes nigrum) oils for wound healing applications. Gellan gum (GG) and kappa-carrageenan (KC) were also used as stabilizers. In order to achieve this, various formulations of the aforementioned substances were performed. It was found that this can offer possible treatment for wound treatments.

The introduction is very short and lacks substance. Are there any related studies to this that should be included? For example, upon cursory search, there are important studies that could be useful for this study including the following below:

1.      https://onlinelibrary.wiley.com/doi/10.1002/ptr.6524

2.      Alves, Anselmo Queiroz MD; da Silva, Valdemiro Amaro Jr PhD; Góes, Alexandre José Silva PhD; Silva, Mariza Severina; de Oliveira, Gibson Gomes PhD; Bastos, Isla Vanessa Gomes Alves PhD; de Castro Neto, Antonio Gomes PhD; Alves, Antonio José PhD. The Fatty Acid Composition of Vegetable Oils and Their Potential Use in Wound Care. Advances in Skin & Wound Care 32(8):p 1-8, August 2019. | DOI: 10.1097/01.ASW.0000557832.86268.64

3.      https://doi.org/10.1590/1414-431X20188209

4.      Guidoni, M., Figueira, M.M., Ribeiro, G.P. et al. Development and evaluation of a vegetable oil blend formulation for cutaneous wound healing. Arch Dermatol Res 311, 443–452 (2019). https://doi.org/10.1007/s00403-019-01919-8

5.      SciELO - Brazil - Production of vegetable oil blends and structured lipids and their effect on wound healing Production of vegetable oil blends and structured lipids and their effect on wound healing

The rationale and need for this study should be substantiated. It lacks compelling evidence on why this type of study should be conducted.

Please provide a flowchart diagram of the procedures as one of the figures.

Remove the word ‘kindly’ in section 2.1.

I think the manuscript should be written in a clearer manner. As it is now, it is somewhat confusing. It should be very specific. For example, under section 2.2, it stated ‘The formulations were produced by emulsifying two phases…’ This statement does not say anything. What formulations?

How were the formulations generated?

It is unclear as to how many formulations were generated. Was any design of experiments performed in the study?

The parts related for each subsection are unclear. That is, the rationale is unclear especially under the materials and methods section.

For each figure, clarify or provide in the captions the important keypoints or summary of what each figures tells us.

Would these results be safe for human skin? I understand that this was tested in cell lines but would these conditions be similar to that of the wound healing process of the human skin?

Overall, this study looks promising but the lack of rationale and clearer details make it not appealing for readers.

Comments on the Quality of English Language

It requires major English language copyediting.

Author Response

REVISOR 2

Polysaccharide-Stabilized Semisolid Emulsion with Vegetable Oils for Skin Wound Healing: Impact of Composition on Physicochemical and Biological Properties.

The overall goal of this study was to utilize avocado (Persea Gratissima) and blackcurrant (Ribes nigrum) oils for wound healing applications. Gellan gum (GG) and kappa-carrageenan (KC) were also used as stabilizers. In order to achieve this, various formulations of the aforementioned substances were performed. It was found that this can offer possible treatment for wound treatments.

Answer: We appreciate the time and attention given to reading our article, as well as the relevant feedback provided. The changes have been made and are highlighted in green. Additionally, the questions raised have been addressed below.

The introduction is very short and lacks substance. Are there any related studies to this that should be included? For example, upon cursory search, there are important studies that could be useful for this study including the following below:

  1. https://onlinelibrary.wiley.com/doi/10.1002/ptr.6524
  2. Alves, Anselmo Queiroz MD; da Silva, Valdemiro Amaro Jr PhD; Góes, Alexandre José Silva PhD; Silva, Mariza Severina; de Oliveira, Gibson Gomes PhD; Bastos, Isla Vanessa Gomes Alves PhD; de Castro Neto, Antonio Gomes PhD; Alves, Antonio José PhD. The Fatty Acid Composition of Vegetable Oils and Their Potential Use in Wound Care. Advances in Skin & Wound Care 32(8):p 1-8, August 2019. | DOI: 10.1097/01.ASW.0000557832.86268.64
  3. https://doi.org/10.1590/1414-431X20188209
  4. Guidoni, M., Figueira, M.M., Ribeiro, G.P. et al.Development and evaluation of a vegetable oil blend formulation for cutaneous wound healing. Arch Dermatol Res 311, 443–452 (2019). https://doi.org/10.1007/s00403-019-01919-8
  5. SciELO - Brazil - Production of vegetable oil blends and structured lipids and their effect on wound healing Production of vegetable oil blends and structured lipids and their effect on wound healing

Answer: Thank you for your valuable suggestion regarding the introduction. We appreciate your observation on the need to provide more substance and context by including relevant studies that emphasize the benefits of vegetable oils for wound healing. In response to your feedback, we have expanded the introduction to include recent and pertinent studies, which highlight the anti-inflammatory, antioxidant, and skin-repairing properties of various vegetable oils and their beneficial impact on the wound healing process. Specifically, we have integrated findings from studies on the fatty acid composition and therapeutic potential of these oils in cutaneous applications, as well as their ability to support skin barrier function and promote tissue regeneration (DOIs: 10.1097/01.ASW.0000557832.86268.64, 10.1007/s00403-019-01919-8, 10.1590/1414-431X20188209, 10.1002/ptr.6524, and 10.1590/S1984-82502015000200019). This enhancement provides a broader scientific basis for the study and supports the rationale for using vegetable oils in combination with polysaccharides for wound care formulations.

We believe these additions significantly strengthen the manuscript and align with your recommendation to offer a more comprehensive background on the topic. For enhanced comprehension, we included in the revised document additional information about the topic, which is highlighted in green (Lines 56 to 79).

The rationale and need for this study should be substantiated. It lacks compelling evidence on why this type of study should be conducted.

Answer: We apologize for our justification not being sufficiently clear and thank you for your insightful comment regarding the need to strengthen the rationale for this study. We agree that providing compelling evidence to support the significance of this work is essential. In response, we have added a paragraph in the introduction that highlights the existing challenges in wound care, particularly the need for biocompatible, natural, and effective formulations capable of addressing inflammation, oxidative stress, and hydration control. We have also referenced recent studies on the beneficial properties of vegetable oils and polysaccharides, which support the development of innovative wound care products with enhanced therapeutic potential, as you kindly suggested. We believe that this addition underscores the relevance of our study and clarifies its contribution to advancing wound care formulations.

            For enhanced comprehension, we included in the revised document additional information about the topic, which is highlighted in green.

Please provide a flowchart diagram of the procedures as one of the figures.

Answer: Thank you for the constructive suggestion to include a flowchart of the procedures. We have added this diagram as Figure 1 (Page 4), accompanied by an explanatory caption to guide readers through each step of the methodology. We believe this addition significantly enhances the clarity and understanding of the experimental workflow, providing a more comprehensive view of our study.

Remove the word ‘kindly’ in section 2.1.

Answer: The word has been removed from the text.

I think the manuscript should be written in a clearer manner. As it is now, it is somewhat confusing. It should be very specific. For example, under section 2.2, it stated ‘The formulations were produced by emulsifying two phases…’ This statement does not say anything. What formulations?

Answer: Thank you for your valuable feedback regarding the clarity of the manuscript. We agree that making the text more specific will greatly enhance its readability. In response, we have revised the manuscript to improve clarity and precision, addressing ambiguities like the example provided in section 2.2 and specifying each formulation process step in detail. We believe this revision significantly improves the quality and understanding of the work, and we appreciate your constructive input in helping us achieve this.

All changes have been highlighted in green in the revised manuscript for easy reference.

How were the formulations generated?

Answer: Thank you for your question regarding the formulation process. Reviewer 1 similarly noted some difficulty in understanding the details of the formulations we prepared. In response, we have revised the methodology section to provide a more detailed and clear description of each formulation step. Additionally, we have included a detailed flowchart outlining the preparation steps, which we believe will enhance clarity and assist readers in following the procedure. We appreciate your feedback, which has contributed to improving the manuscript.

It is unclear as to how many formulations were generated. Was any design of experiments performed in the study?

Answer: Thank you for your question regarding the number of formulations and the use of a design of experiments (DoE). In this study, we prepared four formulations: F1 (gellan gum with avocado oil), F2 (kappa-carrageenan with avocado oil), F3 (gellan gum with blackcurrant oil), and F4 (kappa-carrageenan with blackcurrant oil), including placebo formulations, as described in tables 1 and 2. Rather than utilizing a DoE approach, these formulations were selected based on insights from previous literature to explore the combined effects of these specific polysaccharides and vegetable oils.

We agree that applying a DoE would provide more robust and consistent data on the impact of each component in the formulation. However, our primary objective in this study was to evaluate the feasibility of preparing stable emulsions with these natural components and assess their biological impact on an in vitro wound healing model. Future studies will focus on refining and optimizing the most promising formulation identified here, employing a DoE approach to thoroughly investigate the influence of each variable on the formulation’s performance. We acknowledge that the absence of DoE may be regarded as a potential limitation of our study. This aspect has been briefly addressed in the revised manuscript, which is highlighted in green (Lines 406 to 416).

The parts related for each subsection are unclear. That is, the rationale is unclear especially under the materials and methods section.

AnswerThank you for your observation regarding the clarity and rationale in the materials and methods section. We recognize that providing a clear justification for each methodological step is essential for comprehending the study's structure and purpose. In response, we have revised the materials and methods section to clarify the purpose behind each subsection and the rationale for selecting specific procedures and formulations. We believe these adjustments enhance the logical flow and transparency of our methodology, making it easier for readers to understand the experimental design and its relevance to the study objectives.

All changes have been highlighted in green in the revised manuscript for easy reference.

For each figure, clarify or provide in the captions the important keypoints or summary of what each figures tells us.

Answer: Thank you for your insightful suggestion to enhance the clarity of our figures. We have revised the captions to incorporate key points and a concise summary of the main findings presented in each figure. This additional context improves the interpretation of the data and emphasizes the significance of each figure in relation to the study's overall conclusions. We truly value this feedback, as it has enabled us to create a more reader-friendly presentation of our results.

All changes have been highlighted in green in the revised manuscript for easy reference.

Would these results be safe for human skin? I understand that this was tested in cell lines but would these conditions be similar to that of the wound healing process of the human skin?

Answer: Thank you for your comment. This study primarily focused on in vitro testing using cell lines to evaluate the formulations' biocompatibility and wound-healing potential. However, we recognize that cell line models provide only a limited initial indication of safety and efficacy. The in vitro conditions do not fully capture the complexity of human wound healing, which involves intricate immune responses, diverse cell types, and complex tissue architecture.

Future research will utilize more advanced models, such as ex vivo human skin or animal models, to better replicate human wound healing conditions. These additional evaluations will allow us to assess the formulations’ safety and effectiveness, as well as any potential inflammatory or sensitizing effects in a context that more closely aligns with human skin physiology. We appreciate your question, as it underscores the importance of conducting further testing to confirm the suitability of these formulations for human use. We recognize that the lack of specific investigations to ensure skin safety may be a potential limitation of our study. This issue has been briefly addressed in the revised manuscript and highlighted in green (Lines 820 to 829).

Overall, this study looks promising but the lack of rationale and clearer details make it not appealing for readers.

Answer: We understand the importance of providing a clear rationale and detailed methodology to enhance the study's appeal and accessibility for readers. In response, we have revised the introduction to strengthen the rationale, emphasizing the need for innovative, natural wound care formulations. Additionally, we have clarified the details in the materials and methods section, improving the transparency of each step involved in the study. We believe these adjustments make the manuscript more engaging and comprehensive, better aligning it with the interests and understanding of our readership.

All changes have been highlighted in green in the revised manuscript for easy reference.
